# Enhancing machine learning performance in cardiac surgery ICU: Hyperparameter optimization with metaheuristic algorithm

Ali Bahrami[1], Morteza Rakhshaninejad [1], Rouzbeh Ghousi [1]*, Alireza Atashi[2,3]

1 School of Industrial Engineering, Iran University of Science and Technology, Tehran, Iran, 2 Department of Digital Health, School of Medicine, Tehran University of Medical Sciences, Tehran, Iran, 3 Cancer Informatics Research Group, Clinical Research Department, Breast Cancer Research Center, Motamed Cancer Institute, ACECR, Tehran, Iran

* ghousi@iust.ac.ir

**Data Availability Statement:** All data used in this study have been anonymized by removing personally identifiable information from the samples to ensure anonymity and are available

## Abstract

The healthcare industry is generating a massive volume of data, promising a potential gold-mine of information that can be extracted through machine learning (ML) techniques. The Intensive Care Unit (ICU) stands out as a focal point within hospitals and provides a rich source of data for informative analyses. This study examines the cardiac surgery ICU, where the vital topic of patient ventilation takes center stage. In other words, ventilator-supported breathing is a fundamental need within the ICU, and the limited availability of ventilators in hospitals has become a significant issue. A crucial consideration for healthcare professionals in the ICU is prioritizing patients who require ventilators immediately. To address this issue, we developed a prediction model using four ML and deep learning (DL) models—LDA, CatBoost, Artificial Neural Networks (ANN), and XGBoost—that are combined in an ensemble model. We utilized Simulated Annealing (SA) and Genetic Algorithm (GA) to tune the hyperparameters of the ML models constructing the ensemble. The results showed that our approach enhanced the sensitivity of the tuned ensemble model to 85.84%, which are better than the results of the ensemble model without hyperparameter tuning and those achieved using AutoML model. This significant improvement in model performance underscores the effectiveness of our hybrid approach in prioritizing the need for ventilators among ICU patients.

## 1. Introduction

The development of biomedical equipment and healthcare services has enabled the Intensive Care Unit (ICU) to collect vast amounts of data. This advancement has led to a growing interest in analyzing this data for various purposes, such as improving patient care and predicting patient outcomes [1]. Early and accurate diagnosis is essential for reducing mortality rates. Accurately predicting the risk of death for patients awaiting heart surgery can provide crucial information, enabling life-saving interventions while also reducing costs and time. The

from the figshare database (DOI: https://doi.org/10.6084/m9.figshare.26175826.v1). Alongside the data, all codes utilized in the study are also available in the same figshare repository. Data are available from a hospital associated with Shahid Beheshti University of Medical Sciences and Health Services in Iran. We utilized data from a hospital where patients consent to their information being used for research purposes upon hospitalization, ensuring adherence to ethical considerations.

**Funding:** The author(s) received no specific funding for this work.

**Competing interests:** The authors have declared that no competing interests exist.

immediate implementation of predictive mortality risk assessment is essential for heart surgery patients [2].

Common scoring systems, including APACHE (Acute Physiology and Chronic Health Evaluation) and SAPS (Simplified Acute Physiology Score), are frequently used to predict mortality in ICUs. Although these models are robust in certain contexts, their effectiveness, particularly of the APACHE II system in prolonged mechanical ventilation cases, is still debatable. Despite lingering uncertainty, the APACHE II score has emerged as a reliable predictor of mortality for patients undergoing weaning from prolonged mechanical ventilation [3]. According to Khwannimit et al. [4], a customized version of APACHE II surpasses a customized SAPS II in its accuracy for predicting in-hospital mortality. According to their findings, the tailored APACHE II may be used by ICUs with comparable patient profiles for efficient quality assessment and mortality prediction. Conversely, Poole et al. [5] observed that SAPS II and SAPS III were not good enough at predicting death, with SAPS III being a surprise in overestimating mortality compared to SAPS II. lacked precision in mortality prediction, with SAPS III unexpectedly overestimating mortality.

The integration of machine learning (ML) with traditional scoring systems marks a significant advancement in predicting outcomes like mortality and ventilation requirements, including weaning success and prolonged ventilation. According to Kong et al. [6] research, strong predictive skills were shown by the ML-based techniques developed. Interestingly, the model known as the Gradient Boosting Machine (GBM), proved to be the most effective in predicting the risk of in-hospital death. Advanced ML models, including Balanced Random Forest (BRF), Light Gradient Boosting Machine (LGBM), Extreme Gradient Boosting (XGB), Multilayer Perceptron (MLP), and Logistic Regression (LR), have demonstrated superior accuracy over traditional systems in predicting 30-day mortality for mechanically ventilated patients [7]. Mechanical ventilation, a critical intervention in ICUs, is typically administered through an endotracheal tube, known as invasive mechanical ventilation, to assist patients with severe respiratory problems [8].

One of the primary challenges in ICU care is the process of weaning patients off ventilators. Research has shown that ML techniques, including LR, XGBoost (Extreme Gradient Boosting), and Support Vector Machines (SVM) that utilize related variables, are effective in accurately predicting the outcomes of ventilator weaning [9]. ML models utilizing the XGBoost and CatBoost algorithms have demonstrated brilliant accuracy in predicting the need for mechanical ventilation and assessing the mortality risk in COVID-19 patients [10]. These models stand out for their precision in critical healthcare scenarios, particularly in managing the pandemic's challenges.

In the realm of pediatric cardiac ICU, mechanical ventilation remains a cornerstone of treatment. This is especially true for patients with serious cardiovascular problems, where the weaning process from artificial breathing is a crucial and delicate part of their treatment [11]. Li et al. [12], highlighted the increased mortality risk associated with patients undergoing mechanical ventilation for congestive heart failure (CHF). They developed and validated a CatBoost model capable of accurately predicting hospital mortality in this patient group. Hsieh et al. [13] conducted a comparative study of various ML models against conventional metrics to predict the mortality rate of patients undergoing unscheduled extubation (UE) in ICU. Their findings revealed that the random forest (RF) model was particularly effective in predicting mortality among these patients.

Moreover, Meenen et al. [14] discovered a correlation between the mechanical power of ventilation, the driving pressure, and important patient outcomes like ventilation time and mortality. Their study focused on determining how well these features predicted mortality, particularly 24 hours after invasive breathing started. The development of effective natural ventilation systems for sustainable building designs is becoming more and more important.

Park et al. [15] examined eight ML algorithms to predict natural ventilation rates. The interpretation of nonlinear correlations between environmental variables both indoors and outdoors is appropriate for these methods. Furthermore, Liang et al. [16] proposed an automated model using data from the MIMIC-III cohort to predict ventilator-associated pneumonia (VAP). The model showed impressive performance, with high scores in AUC (Area Under the Curve), sensitivity, and specificity metrics. Sayed et al. [17] attempted to determine the optimum early course of action within the first 48 hours in the ICU. They applied supervised machine learning techniques to estimate the duration of mechanical ventilation needed following the onset of acute respiratory distress syndrome (ARDS). Another study [18] utilized a variety of ML methods to predict the need for early treatments for Multiple Organ Dysfunction Syndrome (MODS) in the ICU, based on data regarding ventilator use.

Several studies have aimed to predict extubation failure, intubation, successful ventilator mode shifting, or prolonged mechanical ventilation in the ICU. These studies have achieved good results [19–24]. Last but not least, Shashikumar et al. [25] developed a transparent DL strategy to predict hospitalized patients' need for mechanical ventilation, including those with COVID-19, using publicly available data from electronic health records.

Bahrami et al [26], presented a two-stage hybrid model based on combination of ML and expert opinion to predict the patients' need for ventilators in the ICU of cardiac surgery and then prioritized patients to assign limited ventilator to critically patients first.

Despite significant advances, few studies have specifically focused on predicting the need for ventilators in cardiac surgery ICU patients. Our study aims to address this gap by employing ML techniques to forecast the necessity of ventilators for these patients. This predictive capability is especially vital during pandemics or periods of increased ICU admissions, where efficient resource allocation can be lifesaving. To enhance the performance and reliability of ML models, it is crucial to conduct hyperparameter tuning. This process is essential for optimizing the functionality of ML algorithms.

The use of metaheuristic algorithms for selecting optimal parameters of ML models is a common and effective approach [27]. Some popular algorithms in this domain include Genetic Algorithms (GA) [28], Simulated Annealing (SA) [29], and Ant Colony Optimization (ACO) [27]. Among all these methods, the SA metaheuristic algorithm for hyperparameter tuning due to its proven effectiveness in navigating complex solution spaces and efficiently identifying optimal parameters in the healthcare systems is better option. This choice is grounded in SA's unique ability to avoid local optima, a common challenge in model optimization, thereby ensuring a more robust and accurate performance of our ML models in critical healthcare settings [29].

We opted for metaheuristic algorithms such as SA and GA over traditional methods like cross-validation and grid search [30] due to their superior efficiency in navigating complex and high-dimensional parameter spaces. These algorithms excel at finding optimal solutions more efficiently than exhaustive search methods such as grid search [31], which becomes computationally impractical as the number of hyperparameters increases. Unlike grid search, metaheuristics can escape local optima and explore the solution space more globally, which is crucial for achieving optimal model performance [32]. Additionally, we reviewed these models as presented in Table 1 of this study.

In This study, to improve our previous work [26] and to improve performance of ML and DL algorithms including linear discriminant analysis (LDA), CatBoost, and ANN while predicting necessity of ventilators for patients, we employed the SA for hyperparameter tuning. This strategy is pivotal in optimizing model parameters, enhancing evaluation metrics, and preventing overfitting, thus improving predictive accuracy in critical healthcare settings.

**Table 1. A summarized list of related research works.**

| Article | Scoring Models: SAPS, APACHE, ... | Used Model | | | | | | | | | Metaheuristic Algorithm | Feature Selection | Hybrid Model |
|---|---|---|---|---|---|---|---|---|---|---|---|---|---|
| | | Machine Learning | | | | | | | Deep Learning | | | | |
| | | Supervised | | | | | | | ANN | | | | |
| | | Classification | | | | | | | | | | | |
| | | GBM | XGBoost | LDA | RF | CATBoost | Regression | Ensemble | | | | | |
| Khwannimit & Bhurayanontachai.[4] | * | | | | | | | | | | | | |
| Poole et al. [5] | * | | | | | | | | | | | | |
| Rojek-Jarmuła et al. [3] | * | | | | | | | | | | | | |
| Hsieh et al. [13] | * | | | | * | | | | * | | | | |
| T. Chen et al. [23] | | | * | | | | | | * | | | * | |
| Ghorbani et al. [1] | * | | * | | * | | | * | * | | | * | * |
| Mansoori et al. [28] | | | | | * | | | * | * | | * | | |
| van Meenen et al. [14] | * | | | | | | | | | | | | |
| Kong et al. [6] | * | * | | | * | | | * | | | | | |
| Jia et al. [8] | | | | | * | | | | * | | | | |
| Kim et al. [7] | * | | * | | * | | | | * | | | | |
| Yu et al. [10] | | | * | | | * | | | | | | | |
| Ahmed et al. [29] | | | | | * | * | | | | | * | * | |
| Park & Park. [15] | | | | | | | | * | * | | | * | |
| Sayed et al. [17] | | | * | | * | | | | | | | | |
| Otaguro et al. [20] | * | | * | | * | | | | | | | | |
| Zhao et al. [22] | * | * | * | | * | * | | | * | | | * | |
| Arvind et al. [24] | | | | | * | | | | | | | | |
| Shashikumar et al. [25] | | | | | | | | | * | | | | |
| Arman Ghavidel et al. [2] | | | * | | * | | | * | | | | * | |
| Ali et al. [27] | | | | | | | | | | | * | * | |
| Rooney et al. [11] | | | | | * | | | | | | | | |
| Cheng et al. [21] | * | | * | | * | | | | * | | | * | |
| Li et al. [12] | * | | * | | | * | * | | * | | | * | |
| W.-T. Chen et al. [9] | | | * | | * | | | | * | | | * | |
| Liang et al. [16] | * | | | | * | | | * | | | | | |
| Liu et al. [18] | | | * | | * | | | * | * | | | * | |
| Vali et al. [19] | * | | | | * | | | | * | | | | |
| Bahrami et al. [26] | | * | * | * | * | * | | | * | | | * | * |
| **This study** | | * | * | * | * | * | | * | * | | * | * | * |

Table 1 presents a summarized overview of research studies focused on predicting mortality and ventilation-related cases in ICU patients. Our study introduces five key innovations:

- A hybrid model combining ML and DL models with metaheuristic algorithm.

- The application of ML and DL models to forecast ventilator needs in the cardiac surgery ICU.

- The use of RF and GBM for calculating the importance of features.

- Hyperparameter tuning with the SA and GA metaheuristic algorithm.

- Formation of an ensemble model using weighted voting to enhance prediction of ventilator needs.

The rest of this study is structured as follows: Section 2 demonstrates information about the dataset and the preprocessing techniques utilized in this study. Also in section 2, we present the Hybrid model and the model evaluation. Section 3 shows the results and discussion, and finally the conclusions of this study and suggestions for future endeavors are outlined in Section 4.

## 2. Materials and methods

An essential component of every research project is comprehending the issue. This study intends to develop a hybrid model to predict hospitalized patients undergoing cardiac surgery ICU's requirement for ventilators. We take this action to avoid the hazards to life and finances associated with delayed ventilator assignment for patients. Thus, in response to this challenge, we present our hybrid model as a potential solution. It's noteworthy that the majority of the coding for this research was implemented in Python version 3.9.13. Also, entirely practical experiment was run on an Intel Core i7 Laptop running at 2.40 GHz with 16GB of RAM. The methodology utilized to achieve the objectives of this study is depicted in Fig 1.

### 2.1. Dataset

For this study, the dataset was meticulously gathered with ethical considerations from all participants at a hospital associated with Shahid Beheshti University of Medical Sciences and Health Services in Iran [2], and written consent was also obtained from all participants. The data collection, undertaken with due regard for ethical standards under the ethical code IR. ACER.IBCRC.REC.1394.71, was accessed for research purposes on March 7, 2016. The data, which focuses on patients who were admitted to the ICU, includes data obtained both during and after heart surgery procedures. The dataset contains 1,098 records and 32 features, of which 31 are critical for predicting ventilator use, with the "On_pump" feature designated as

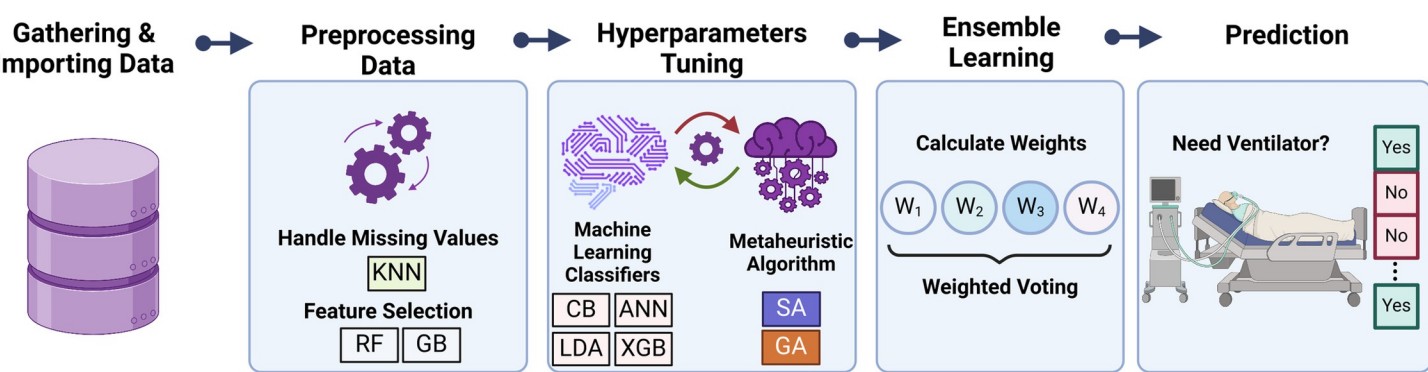

**Fig 1. Different steps of the proposed methodology.**

the key response variable. Table 2 reveals the main features of the dataset in detail. The subsequent section will elaborate on the data preprocessing methods employed in this study.

## 2.2. Preprocessing

Data preprocessing is a crucial and indispensable step in enhancing the development of ML models. Data preparation, the third phase of the CRISP-DM framework, is a critical stage of data mining projects, consuming a considerable portion of the overall time and effort. Thus, the quality and nature of the data are paramount for the success of the ML modeling process. To optimize the effectiveness of our modeling, we undertake five detailed steps of data preprocessing, which are crucial for the integrity of our analysis. The following are the steps we will implement in this section:

**Table 2. Main features of our dataset of cardiac surgery ICUs.**

| Feature name | Description | Type |
|---|---|---|
| Code | ID for each patient. | Numeric |
| Age | Age for each patient. | Numeric |
| Sex | Gender for each patient. | Nominal |
| Weight | Weight for each patient. | Numeric |
| Hight | Hight for each patient. | Numeric |
| Cross_clamp | Generation of heart rate for the patients. (0,1 for No,Yes) | Numeric |
| Arrythmia | Abnormality of the patient's heart rhythm. (0,1 for No,Yes) | Numeric |
| Number_of_Grafts | Number of transplants that the patient has done. | Numeric |
| CABG_Valve | Patient's condition regardless of whether the patient has undergone coronary artery bypass graft surgery. (0,1 for No,Yes) | Numeric |
| Valve | Patient's condition regardless of whether the patient has undergone coronary artery bypass graft surgery. (No or Yes) | Nominal |
| IABP | Patient's condition regardless of whether the patient has IABP or not. (0,1 for No,Yes) | Numeric |
| MI | Patient's condition regardless of whether or not the patient has myocardial infarction. (0,1 for No,Yes) | Numeric |
| HTN | Patient's condition regardless of whether or not the patient suffers from hypertension. (0,1 for No,Yes) | Numeric |
| HLP | Patient's condition regardless of whether or not the patient has hyperlipidemia. (0,1 for No,Yes) | Numeric |
| COPD | Patient's condition whether or not the patient has chronic obstructive pulmonary disease. (0,1 for No,Yes) | Numeric |
| Diabetes | Patient's condition whether or not the patient has diabetes. (0,1 for No,Yes) | Numeric |
| Renalfailure | Patient's condition whether or not the patient has renalfailure. (0,1 for No,Yes) | Numeric |
| PreviousAF | Patient's condition regardless of whether or not the patient has already undergone cardiac surgery. (0,1 for No,Yes) | Numeric |
| Anemia | Patient's condition whether or not the patient has anemia. (0,1 for No,Yes) | Numeric |
| Smoker | Patient's condition whether or not the patient is a smoker. (0,1 for No,Yes) | Numeric |
| Addiction | Patient's condition whether or not the patient has addiction. (0,1 for No,Yes) | Numeric |
| Grade_of_Age | Specific grade of age for each patient. (1,2,3,4 for age of 0–25, 26–50, 51–75, 50–100) | Numeric |
| EF | Rate of the ejection fraction of the heart. | Numeric |
| Albumin | Condition of patient whether or not the patient received albumin. (0,1 for No,Yes) | Numeric |
| MAP | Mean arterial pressure throughout one cardiac cycle, systole, and diastole. | Numeric |
| Arrest | Condition of patient whether or not the patient had a cardiac arrest while he was hospitalized. (0,1 for No,Yes) | Numeric |
| PH | Blood's ph for each patient. | Numeric |
| MG | Blood's magnesium for each patient. | Numeric |
| Duration | Duration of each patient's stay in the intensive care unit. | Numeric |
| CHF | Patient's condition whether or not the patient has chronic heart failure. (0,1 for No,Yes) | Numeric |
| Off_pump | Condition of patients whether if ventilator is connected to the patient or not. (1,0 for No,Yes) | Numeric |
| On_pump | Condition of patients whether if ventilator is connected to the patient or not. (0,1 for No,Yes) | Numeric |

[1] Source of data: cardiac surgery ICU of hospital related to Shahid Beheshti University of Medical Sciences and Health Services in Iran.

- Handling missing values and applying feature importance

- Utilizing one-hot encoding for categorical variable

- Partitioning the data into train, validation, and test sets

- Improving imbalanced dataset by using SMOTE method

- Using the Z-Score in order to standardize the data

**2.2.1. Handling missing values and applying feature importance.** Our initial dataset comprised 32 columns (31 independent features and 1 dependent feature), as detailed in Table 2. The presence of null values does not necessarily indicate data missingness; for instance, null values in binary columns often represent the value 0. To ensure accurate interpretation, we consulted a medical expert familiar with the data collection process. Several features were identified as redundant or non-essential and were subsequently removed to refine the dataset. The 'Code' feature, representing patient IDs, was removed as it did not contribute to our analysis. Similarly, the 'Valve' feature was removed as it duplicated information found in the 'CABG_Valve' feature, and the 'Off_pump' feature, having values opposite to those of 'On_pump', was also eliminated.

Furthermore, certain features exhibited a high percentage of missing values, prompting specific actions: 'DURATION' and 'CHF', with more than 95% missing values, were removed to preserve the validity and reliability of our model. For features 'PH', 'MG', and 'MAP', which had more than 15% but less than 50% missing values, we employed the K-Nearest Neighbors (KNN) imputation method. This method was crucial in preserving the underlying data structure and maintaining feature integrity by leveraging the similarity between data points. Additionally, six samples that lacked any values were removed.

Upon addressing these issues, 26 of the initial 31 independent features and 1092 of the 1098 samples were retained for further analysis. We also conducted a feature importance analysis using both RF and GBM models, as illustrated in Fig 2. This analysis was instrumental in identifying and retaining the most informative features, using a threshold importance score of 0.001. Only features with significant predictive power were included in further analyses. The intersection of important features identified by these models, as shown in Fig 2, included 'Arrythmia', 'Number_of_Grafts', 'Addiction', 'Arrest', 'Hight', 'Weight', 'MAP', 'MG', 'Grade_of_Age', 'Age', 'CABG_Valve', 'EF', 'IABP', 'Cross_Clamp', 'Smoker', 'MI', 'Albumin'. These features were carefully considered to ensure robustness in our feature set.

**2.2.2. Utilizing one-hot encoding.** To incorporate categorical variables into ML models, it's advisable to transform them into binary variables using one-hot encoding [10, 22]. As an instance, a categorical feature like sex, which can only be female or male, cannot be directly incorporated into a ML model; This technique transforms the Sex column into a binary column, where male is represented by 1 and female by 0.

**2.2.3. Partitioning the data into train, validation, and test sets.** To mitigate the risk of overfitting in ML models, we split the dataset into separate training, validation and testing sets [7]. The primary objective of this data partitioning is to evaluate the model's performance on unseen data, thereby ensuring its generalizability. The training set is used initially to develop the models, while the validation set is applied for fine-tuning the hyperparameters of these models using metaheuristic algorithms. Finally, we evaluate their performance on the test set, which provides insights into their general applicability. To determine the most effective model, we evaluated their performance on various metrics. To achieve this, we divided the data into a 60-20-20 split, with 60% dedicated to training, 20% reserved for validation, and

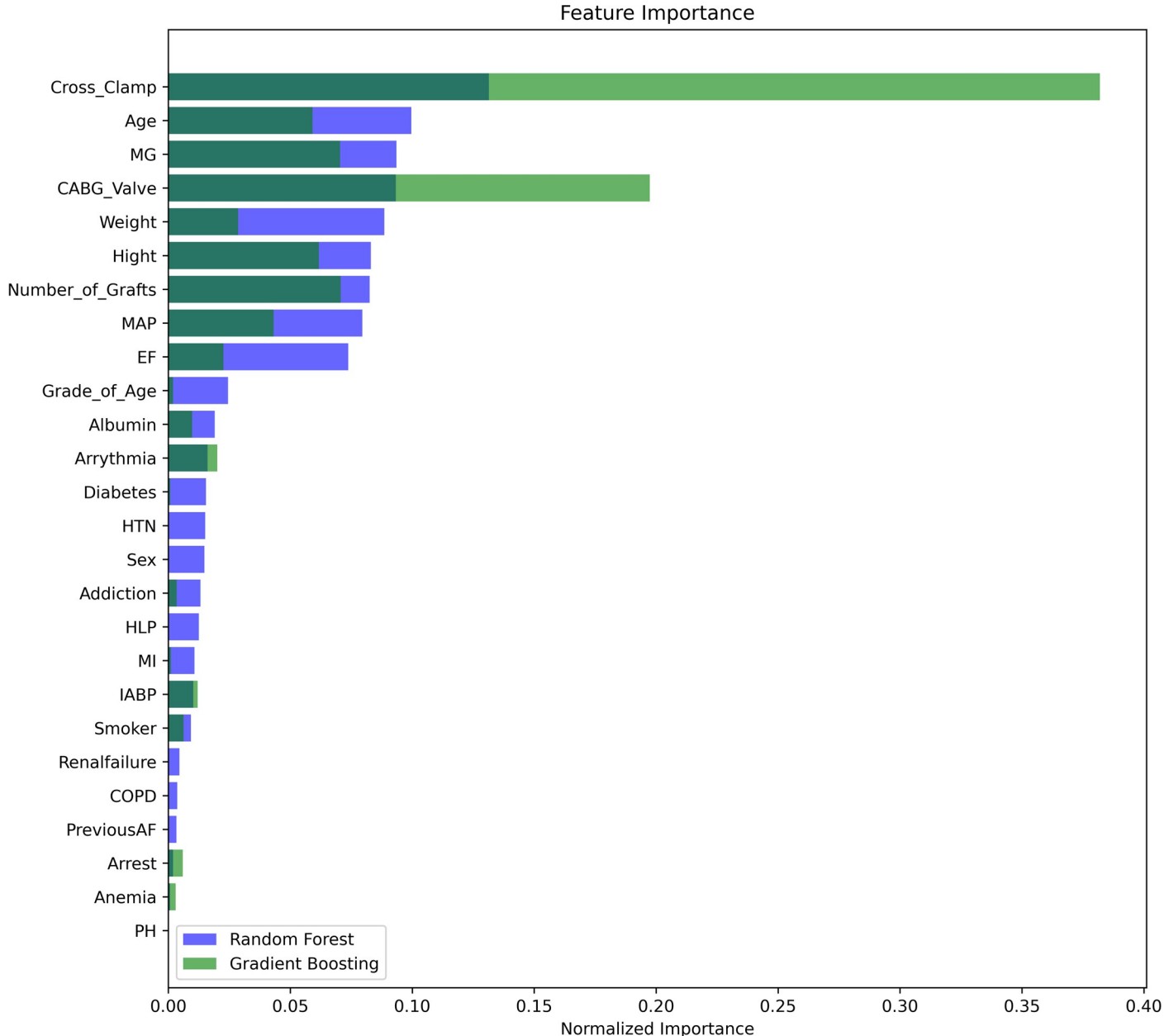

**Fig 2. Feature importance with Random Forest and Gradient Boosting.**

20% for testing. Specifically, of the 1092 total records, 655 were utilized to train the models, 218 for validation, and 219 to test their performance. This methodical approach allows for a comprehensive assessment of each model's ability to generalize beyond the training data.

**2.2.4. Improving imbalanced data by using SMOTE method.** To address potential biases introduced by imbalanced data in our study, we employed the Synthetic Minority Over-sampling Technique (SMOTE). This method is particularly effective in balancing datasets by synthesizing new samples for the under-represented class [1]. By using SMOTE, we enhanced the dataset's diversity and improved the generalizability of our models. SMOTE works by selecting data points that are close in the feature space, drawing a line between the points in the feature

space, and creating a new point along that line. This approach is crucial for creating a balanced dataset and avoiding model bias towards the majority class.

SMOTE was applied exclusively to the training data to balance it. Initially, the training data had an imbalance between the two classes: class 0 (patients who do not need a ventilator) and class 1 (patients who need a ventilator). After applying SMOTE to the training data, we achieved a balanced distribution with 414 samples in each class, totaling 828 samples. In contrast, the validation set, which did not undergo SMOTE, contains 142 samples that do not need a ventilator and 76 that do, reflecting the original distribution of the dataset. Similarly, the test set has 154 samples that do not need a ventilator and 65 that do. This strategic application of SMOTE ensures that our model training is robust and fair while allowing us to assess model performance against the more naturally imbalanced conditions seen in the validation and test data.

**2.2.5. Using the Z-score to standardize the data.**   Feature scaling or standardization, a critical step in data preprocessing, is essential when dealing with datasets where features have varying scales or are measured in distinct units. The discrepancies in feature ranges can hinder the effectiveness of many ML models. For instance, in algorithms that use distance-based calculations, a feature with a significantly larger range will exert an undue influence on the distance calculations.

The Z-score is a popular approach for standardizing data. It entails subtracting the mean of each feature and then dividing by its standard deviation [1].

$$Z = \frac{(x - \mu)}{\sigma} \tag{1}$$

The Z-scores for each sample are calculated in Eq (1) as you can see above.

Upon completing the standardization procedure, all features will be centered at zero and have a standard deviation of one, ensuring a consistent scale. This standardization is performed separately for the train, validation, and test sets. Standardizing these sets separately safeguards against data leakage, a phenomenon where information from the validation and test data infiltrates the modeling process.

## 2.3. Modeling

Previously, we highlighted the significance of patient ventilation in the ICU setting. Mechanical ventilation, which provides artificial respiration, is a crucial requirement for ICU patients, particularly in light of the limited availability of ventilators within hospitals. Healthcare workers strive to distinguish patients who require ventilators with the highest level of urgency. To address this challenge, we developed a hybrid ensemble predictive model to predict the need for ventilator among patients in the cardiac surgery ICU.

To address the critical need for ventilators in cardiac surgery ICU and prevent life-threatening consequences due to delayed ventilator allocation, we propose our ensemble model to accurately predict ventilator requirements as below:

**2.3.1. Machine learning & metaheuristic models.**   In the first stage of our ensemble model, we implemented four classification models to predict the necessity of ventilators for patients in the cardiac surgery ICU. This involved the use of supervised ML and DL models, specifically LDA, CatBoost, ANN, and XGBoost, as provided in the Scikit-learn, CatBoost, Keras, and xgboost libraries of Python. Each of these classifiers was trained using the training dataset to assess their performance differences.

We chose CatBoost for its ability to handle categorical features without extensive preprocessing and its efficiency in both regression and classification tasks [33]. LDA was selected for

its strength in dimensionality reduction and class distinction based on linear combinations of features, which is vital for medical datasets [34]. ANN was chosen for its capability to model non-linear relationships in medical data through deep learning architectures [35]. XGBoost was included due to its proven efficacy and recommendation by the TPOT AutoML algorithm [36], enhancing the ensemble's performance. These models were validated against empirical evidence, ensuring their superior performance in metrics such as Accuracy, Precision, Sensitivity, Specificity, and F1-score, as documented in [26] study. We reviewed related studies, as presented in Table 1, to support our model selection.

After this, we need to do hyperparameter tuning to determine which parameters of our classifiers are good to be used in the ensemble model to either improve evaluation metrics or avoid overfitting. Hyperparameters in ML are parameters that values are predetermined before the learning process starts [12]. To optimize these parameters, we employed the SA metaheuristic algorithm, a widely recognized method for its efficiency in navigating and optimizing complex parameter spaces. This approach was chosen for its ability to effectively balance the exploration and exploitation of the parameter space, thereby improving evaluation metrics and reducing the risk of overfitting.

The ANN model underwent parameter adjustments including the number of layers, units per layer, activation functions, optimizer type, loss function, batch size, and epochs. For the LDA model, parameters like solver type, number of components, store covariance option, tolerance level, and shrinkage were fine-tuned. The CatBoost model's tuning focused on parameters such as learning rate, tree depth, L2 leaf regularization, and the number of iterations. **The tuning of the XGBoost model included hyperparameters such as C, dual, loss, penalty, tol, learning rate, max depth, max features, min samples leaf, min samples split, n_estimators, and subsample**. Each of these hyperparameters was carefully selected based on recommendations from developer documentation and insights from related studies, leading to different numbers of parameters being optimized for each model. This variance is inevitable due to the unique characteristics and capabilities of each model, ensuring each is optimized to enhance performance and robustness against overfitting.

To effectively manage potential overfitting in predictive models, we employed a range of regularization techniques tailored to each model's architecture. For the LDA model, we considered a shrinkage parameter to be optimized through SA and GA, effectively balancing model complexity and generalization. In CatBoost model, instead of traditional L1 or L2 regularization, we adjusted the l2_leaf_reg parameter to 3, based on preliminary tests which suggested this setting optimally prevents overfitting while maintaining model flexibility. For XGBoost, we applied L1 regularization with an alpha value of 0.01 to encourage feature selection and L2 regularization with a lambda value of 1.0 to penalize larger weights, thus enhancing model generalization across various datasets. Lastly, in the ANN, both L1 and L2 regularizations were set at 0.01 directly in the layers, and a combined L1_L2 regularization was employed to ensure a balance between reducing model complexity and retaining predictive accuracy.

These optimizations were captured in detailed logs, showing the best accuracy, precision, recall (Sensitivity), F1 score, specificity.

The hyperparameters set for each ML and DL model are summarized in Table 3. This table provides a comprehensive overview of the hyperparameters utilized during the tuning phase. The ranges for these hyperparameters were determined based on the developer documentation for each model, expert feedback tailored to our problem characteristics, and insights from other studies, which are referenced in the last column of Table 3. This approach ensures that the tuning is grounded in both theoretical best practices and practical insights relevant to our

**Table 3. The hyperparameters set for each ML and DL model.**

| Model | Parameters | Description / Range of Values | Refrences |
|---|---|---|---|
| **ANN** | Number of Layers | 1, 2, 3, 4, 5, 6, 7, 8, 9, 10, 12, 15 | [1, 37–46] |
| | Units per Layer | 8, 16, 32, 64, 128, 256, 512, 1024 | |
| | Activation Function | relu, sigmoid, tanh, elu, softmax, softplus, selu, softsign. | |
| | Optimizer | adam, sgd, rmsprop, adamax, nadam, adagrad, adadelta, ftrl. | |
| | Loss Function | binary_crossentropy, hinge, squared_hinge, logcosh. | |
| | Batch Size | 8, 16, 32, 64, 128, 256, 512 | |
| | Epochs | 10, 30, 50, 75, 100, 150, 200, 300, 500 | |
| **LDA** | Solver | svd, lsqr, eigen | [26, 47–50] |
| | Number of Components | 0, 1 | |
| | Store Covariance | True, False | |
| | Tolerance | 1.0e-5 to 1.0e-1 | |
| | Shrinkage | None, auto, 0.0–1.0 | |
| **CatBoost** | Learning Rate | 0.01, 0.03, 0.05, 0.07, 0.1, 0.15, 0.2 | [26, 51–59] |
| | Depth | 4, 6, 8, 10, 12, 14, 16 | |
| | L2 Leaf Reg | 1, 3, 5, 7, 9, 12, 15 | |
| | Iterations | 50, 100, 150, 200, 300, 400, 500 | |
| **XGBoost** | C | 0.01, 0.1, 1, 10, 100 | [26, 51, 52, 54, 55, 58, 60–63], |
| | Dual | False, True | |
| | Loss | log_loss | |
| | Penalty | l2, none | |
| | Tol | 1e-4, 1e-3, 1e-2 | |
| | Learning_Rate | 0.01, 0.05, 0.1, 0.2 | |
| | Max_Depth | 3, 4, 5, 6, 7, 8, 9, 10 | |
| | Max_Features | None, sqrt, log2 | |
| | Min_Samples_Leaf | 1, 2, 4 | |
| | Min_Samples_Split | 2, 5, 10 | |
| | N_Estimators | 100, 200, 300, 400, 500 | |
| | Subsample | 0.5, 0.7, 1.0 | |

specific research context. Also, the below pseudo code (Algorithm 1) represents the process of hyperparameter tuning using SA:

**Algorithm 1. Pseudocode for Hyperparameter Selection using Simulated Annealing (SA)**

```
1. Define the hyperparameter space for each model:
    ANN_hyperparameters = [number_of_layers, units_per_layer, activa-
tion_functions, optimizer, loss_function, batch_size, epochs]
    LDA_hyperparameters = [solver, number_of_components, store_covar-
iance, tolerance, shrinkage]
    CatBoost_hyperparameters = [learning_rate, depth, l2_leaf_reg,
iterations]
    XGBoost = [C, dual, loss, penalty, tol, learning_rate, max_depth,
max_features, min_samples_leaf, min_samples_split, n_estimators, and
subsample]
2. Initialize the SA algorithm parameters:
    -initial_temperature (set initial temperature for SA) = 100
    -cooling_rate (set cooling rate for the SA process) = 0.95
    -max_iterations (set maximum number of iterations for SA) = 250
3. For each model (ANN, LDA, CatBoost, XGBoost):
    -Initialize best_solution and best_score to None or initial values
```

```
        -Set current_solution to a random selection from the model's hyper-
parameter space
        -Evaluate current_solution using the model's evaluation function
(e.g., evaluate_ann for ANN)
        -Set best_solution to current_solution if it's the first iteration
or better than the existing best_solution
    For iteration in range(max_iterations):
        -Generate new_solution by slightly modifying current_solution's
parameters
        -Evaluate new_solution using the model's evaluation function
        -Calculate acceptance_probability using SA algorithm's accep-
tance criteria
        -If acceptance_probability > random value between 0 and 1:
        -Update current_solution to new_solution
        -Update best_solution if new_solution is better than
current_solution
        -Reduce temperature based on cooling_rate
    -Return final best_solution for the model
 4. After completing SA for all models, select the best hyperpara-
meters for each model.
```

To evaluate the predictive performance of the models, we need to measure their performance on the validation set for further tuning and the test set for final evaluation and comparisons. The **ensemble model, constructed from the individually optimized models**, is then tested to choose the best setup based on various evaluation metrics. We will discuss on ensemble model and various evaluation metrics in the next subsection.

**2.3.2. Ensemble model construction.** In our study, we have developed an ensemble model that synthesizes the predictive power of four distinct base classifiers—CatBoost, LDA, ANN and XGBoost—each individually fine-tuned using metaheuristic optimization algorithms. This tuning process was essential for adjusting the hyperparameters specific to our data, maximizing the effectiveness of each model within the ensemble framework.

Our ensemble employs a weighted voting mechanism to integrate outputs from each base model. The weighting system is designed to allocate more influence to models that demonstrate superior performance on the validation set, enhancing the predictive accuracy and reliability of the ensemble's overall output. This methodological choice is particularly effective in leveraging the distinct strengths of each classifier, thereby improving the ensemble's capability to predict ventilator needs accurately.

The model training was conducted on 60% of our dataset, with the remainder split equally between validation and testing. The validation phase was crucial not only for hyperparameter tuning but also for determining the appropriate weights for each model within the ensemble. This setup ensures that our model is robust and well-adapted to the nuances of our dataset, minimizing the risk of overfitting.

Following the training and validation, we evaluated the ensemble model on the test set. This evaluation helped us assess the practical effectiveness of the ensemble in a controlled, yet realistic setting. We calculated key performance metrics such as accuracy, precision, sensitivity, specificity, and F1-score for each component model as well as for the ensemble as a whole. These metrics provided a comprehensive view of how each model contributes to the ensemble and how effectively the ensemble performs as a unit. The final prediction $\hat{y}$ can be represented as:

$$\hat{y} = \text{sign}\left(\sum\nolimits_{i=1}^{n} w_i \cdot f_i(x)\right) \tag{2}$$

Where the parameter *n* represents the number of classifiers integrated within the ensemble. Each classifier, indexed by *i*, contributes to the final prediction with a specific weight $w_i$, which is indicative of its importance or performance relative to the ensemble. The function $f_i(x)$ denotes the prediction output by the *i*-th classifier for the input *x*. Additionally, the function *sign*() is utilized to convert the aggregated weighted sum of predictions into a definitive class label. This conversion is dependent on a threshold, 0.5, which is commonly used in binary classification scenarios to determine the class labels. This ensemble not only capitalizes on the individual strengths of each model but also significantly enhances our ability to make accurate predictions about ventilator needs, which is critical for efficient ICU management and patient care.

**2.3.3. Comparison.** In this study, we implemented the Genetic Algorithm (GA) alongside Simulated Annealing (SA) to conduct an in-depth comparison of these hyperparameter optimization techniques. Our goal was to evaluate their efficacy in tuning the hyperparameters of our base classifiers—LDA, CatBoost, Artificial Neural Network (ANN), and XGBoost. We utilized GA, with hyperparameters defined in Table 3, to find optimal parameter sets for each base classifier. This approach allowed us to directly compare the effectiveness of GA and SA in enhancing the performance of these models by adjusting their hyperparameters to achieve the best possible outcomes. The below is pseudo code (Algorithm 2) of hyperparameter tuning using GA:

**Algorithm 2**. **Pseudocode for Hyperparameter Selection using Genetic Algorithm (GA)**

```
1. Define the hyperparameter space for each model:
  ANN_hyperparameters = [number_of_layers, units_per_layer, activa-
tion_functions, optimizer, loss_function, batch_size, epochs]
  LDA_hyperparameters = [solver, number_of_components, store_covar-
iance, tolerance, shrinkage]
  CatBoost_hyperparameters = [learning_rate, depth, l2_leaf_reg,
iterations]
  XGBoost = [C, dual, loss, penalty, tol, learning_rate, max_depth,
max_features, min_samples_leaf, min_samples_split, n_estimators, and
subsample]
2. For each model (ANN, LDA, CatBoost, XGBoost):
  - Generate an initial population of 20 individuals.
  - For each individual in the population:
   - Calculate the model's performance (F1-score) on validation set.
  - For 250 generations:
   - Selection:
    - Select the top 10 performing individuals from the population to
serve as parents for the next generation.
   - Crossover:
    - For each pair of parents, perform a two-point crossover to pro-
duce offspring. This involves selecting two random crossover points in
the hyperparameter list and swapping the segments between these points
to create new offspring.
   - Mutation:
    - Apply mutations with a 0.1 mutation rate to the offspring. Ran-
domly alter one or more hyperparameters within their defined ranges.
The standard mutation involves randomly selecting a hyperparameter and
changing it to another valid value from its range.
   - Evaluate new generation:
    - Assess the fitness of each new individual in the population
using F1-score.
   - Select the best individual:
```

```
        - After all generations have been processed, identify the indi-
vidual with the highest fitness score as possessing the optimal set of
hyperparameters.
3. Output the best hyperparameter set:
        - Return the best hyperparameters and their associated perfor-
mance metrics.
```

We assessed the performance of each base classifier when their hyperparameters were not tuned using SA and GA, as well as the performance of an ensemble model composed of these untuned classifiers. These initial results are presented in Table 4. Further, we analyzed the performance of each base classifier after their hyperparameters were tuned with both GA and SA. Additionally, we evaluated an ensemble model that was comprised of classifiers tuned with these methods. This allowed us to directly compare the impact of hyperparameter tuning on the performance of individual classifiers and their collective performance within an ensemble framework. The results of this analysis are detailed in Table 6, showcasing the effectiveness of each tuning method.

To provide a broader perspective and benchmark our results against current automated techniques, we incorporated AutoML that named TPOT into our study [36]. This AutoML simplifies the selection of ML models and hyperparameter tuning but typically limits this tuning to a narrow range of parameters. We conducted a detailed comparison between the performance of the AutoML-selected model and our ensemble models, which were tuned using SA and GA, and also evaluated them in their untuned states. This analysis, displayed in Table 7, highlights the performance distinctions and demonstrates how our tuning approaches potentially offer more robust customization compared to AutoML's more generic methodology.

While AutoML requires less technical knowledge and provides a streamlined, efficient process suitable for general applications, it lacks the granular control over hyperparameter settings that SA and GA provide. This control is crucial for addressing the specific needs of complex datasets, like ours, where flexibility in the tuning process is essential. The limited hyperparameter tuning range of AutoML compared to our extensive range may also impact the depth of model optimization achievable. Our study discusses these differences, emphasizing the trade-offs between the ease of AutoML and the detailed control offered by SA and GA, which can significantly influence model performance and transparency in research settings.

This comprehensive evaluation strategy not only highlighted the strengths and weaknesses of hyperparameter tuning methods like GA and SA but also underscored the potential of AutoML as a viable alternative in scenarios where manual tuning may be impractical. By comparing these methods across a range of performance metrics, we gained valuable insights into their applicability and effectiveness in a critical healthcare setting, providing essential guidance for future implementations of ML technologies where prediction accuracy and model reliability are crucial.

**2.3.4. Evaluation.** We utilized five classification metrics to evaluate the predictive capabilities of the developed ensemble, ML and DL models: accuracy, precision, recall (sensitivity),

Table 4. Evaluation metrics of ML and DL models before hyperparameter tuning using SA algorithm.

| Model | Accuracy | Precision | Sensitivity | Specificity | F1-score |
|---|---|---|---|---|---|
| CatBoost | 0.778169 | 0.788321 | 0.760563 | 0.795775 | 0.774194 |
| LDA | 0.686620 | 0.673203 | 0.725352 | 0.647887 | 0.698305 |
| ANN | 0.700704 | 0.705036 | 0.690141 | 0.711268 | 0.697509 |
| XGBoost | 0.802817 | 0.798611 | 0.809859 | 0.795775 | 0.804196 |
| **Ensemble** | **0.816901** | **0.816901** | **0.816901** | **0.816901** | **0.816901** |

**Table 5. ML and DL models with their specific hyperparameters' settings.**

| SA | | | GA | | |
|---|---|---|---|---|---|
| Model | Parameters | Optimized Values | Model | Parameters | Optimized Values |
| ANN | Number of Layers | 8 | ANN | Number of Layers | 6 |
| | Units per Layer | 32, 16, 256, 64, 128, 512, 8, 8 | | Units per Layer | 512, 64, 16, 64, 16, 32 |
| | Activation Function | elu, relu, softsign, elu, elu, tanh, tanh, sigmoid | | Activation Function | relu, relu, selu, softsign, elu, selu |
| | Optimizer | nadam | | Optimizer | adagrad |
| | Loss Function | hinge | | Loss Function | binary_crossentropy |
| | Batch Size | 8 | | Batch Size | 10 |
| | Epochs | 200 | | Epochs | 50 |
| LDA | Solver | eigen | LDA | Solver | eigen |
| | Number of Components | 1 | | Number of Components | 1 |
| | Store Covariance | False | | Store Covariance | False |
| | Tolerance | 0.001 | | Tolerance | 0.0001 |
| | Shrinkage | 0.6 | | Shrinkage | 0.6 |
| CatBoost | Learning Rate | 0.15 | CatBoost | Learning Rate | 0.1 |
| | Depth | 10 | | Depth | 16 |
| | L2 Leaf Reg | 3 | | L2 Leaf Reg | 3 |
| | Iterations | 200 | | Iterations | 500 |
| XGBoost | C | 0.1 | XGBoost | C | 0.01 |
| | Dual | False | | Dual | loss |
| | Loss | log_loss | | Loss | log_loss |
| | Penalty | none | | Penalty | none |
| | Tol | 0.01 | | Tol | 0.0001 |
| | Learning_Rate | 0.05 | | Learning_Rate | 0.2 |
| | Max_Depth | 8 | | Max_Depth | 6 |
| | Max_Features | log2 | | Max_Features | log2 |
| | Min_Samples_Leaf | 1 | | Min_Samples_Leaf | 2 |
| | Min_Samples_Split | 2 | | Min_Samples_Split | 10 |
| | N_Estimators | 400 | | N_Estimators3 | 500 |
| | Subsample | 0.7 | | Subsample | 0.5 |

specificity, and F1-score. To ensure the robustness and reliability of our evaluation, we implemented 10-fold cross-validation at each step of performance assessment. This methodological

**Table 6. Evaluation metrics of ML and DL models after hyperparameter tuning using SA algorithm.**

| Model | MA | Accuracy | Precision | Sensitivity | Specificity | F1-score |
|---|---|---|---|---|---|---|
| CatBoost | SA | 0.778169 | 0.768707 | 0.795775 | 0.760563 | 0.782007 |
| | GA | 0.774648 | 0.770833 | 0.781690 | 0.767606 | 0.776224 |
| LDA | SA | 0.714789 | 0.716312 | 0.711268 | 0.718310 | 0.713781 |
| | GA | 0.714789 | 0.716312 | 0.711268 | 0.718310 | 0.713781 |
| ANN | SA | 0.746479 | 0.733333 | 0.774648 | 0.718310 | 0.753425 |
| | GA | 0.739437 | 0.753731 | 0.711268 | 0.767606 | 0.731884 |
| XGBoost | SA | 0.813380 | 0.806897 | 0.823944 | 0.802817 | 0.815331 |
| | GA | 0.799296 | 0.797203 | 0.802817 | 0.795775 | 0.800000 |
| **Ensemble** | **SA** | **0.849765** | **0.842593** | **0.858491** | **0.841121** | **0.850467** |
| | **GA** | **0.834507** | **0.841727** | **0.823944** | **0.845070** | **0.832740** |

**Table 7. Comparison of tuned ensemble using SA and GA with untuned ensemble and AutoML.**

| Prediction Model | Accuracy | Precision | Sensitivity | Specificity | F1-score |
|---|---|---|---|---|---|
| Untuned Ensemble | 0.816901 | 0.816901 | 0.816901 | 0.816901 | 0.816901 |
| AutoML | 0.8046 | 0.8100 | 0.7958 | 0.8134 | 0.8028 |
| SA Tuned Ensemble | **0.849765** | **0.842593** | **0.858491** | **0.841121** | **0.850467** |
| GA Tuned Ensemble | **0.834507** | **0.841727** | **0.823944** | **0.845070** | **0.832740** |

approach allows us to generate more stable and generalizable performance estimates by averaging results across different subsets of the data.

Accuracy, the ratio of the number of correct predictions to the total number of predictions [8]. This assessment utilizes True Positives (TP), True Negatives (TN), False Positives (FP), and False Negatives (FN) to calculate the model's accuracy, which is defined as follows:

$$\text{Accuracy} = \frac{TN + TP}{TN + FP + FN + TP} \tag{3}$$

The formula for precision represents the proportion of positive predictions that were indeed correct.

$$\text{Precision} = \frac{TP}{TP + FP} \tag{4}$$

The recall (sensitivity) formula is given below. It represents the percentage of actual positive cases that were correctly identified.

$$\text{Sensitivity} = \frac{TP}{TP + FN} \tag{5}$$

The specificity formula is provided below. It indicates the proportion of actual negative cases that were correctly classified.

$$\text{Specificity} = \frac{TN}{TN + FP} \tag{6}$$

The trade-off relationship between precision and recall can result in a model performing well on one metric but poorly on the other. The F1-score addresses this issue by considering both metrics simultaneously [8]. It is presented as below:

$$\text{F1} - \text{Score} = \frac{2}{\frac{1}{\text{Precision}} + \frac{1}{\text{Sensitivity}}} \tag{7}$$

The sensitivity index stands out as a key evaluation tool in our research due to its emphasis on the FN error, which is a critical factor given the importance of ensuring that patients who need ventilators are accurately identified. To avoid putting patients' lives at risk due to inaccurate predictions, we aim to minimize the FN error and, consequently, maximize the sensitivity of our model.

Evaluation metrics of ML and DL models before and after using SA and GA algorithms are represented in Tables 4 and 6 respectively, all calculated using 10-fold cross-validation. As you can see in these tables, the best model based on its metrics is ensemble model tuned with SA.

So, based on our comprehensive evaluation of different models, we determined that ensemble with SA achieved the best performance in predicting ventilator requirements for patients in cardiac surgery ICU.

## 3. Results and discussion

To demonstrate the effectiveness of optimization algorithms in enhancing evaluation metrics, we utilized both SA and GA for hyperparameter tuning across our ML and DL models. Table 4 showcases the evaluation metrics of these models before any hyperparameter tuning was applied. Notably, the ensemble model and XGBoost emerged as frontrunners, with the ensemble achieving the highest scores across all metrics—accuracy, precision, sensitivity, specificity, and F1-score all at 0.816901. In contrast, XGBoost displayed strong performance with the highest sensitivity of 0.809859 among the base classifiers and an impressive F1-score of 0.804196. Table 5 subsequently details the optimized hyperparameters for models such as ANN, LDA, CatBoost, and XGBoost, achieved using both SA and GA, setting the stage for a potential enhancement in their respective performance metrics.

Table 6 presents a clear visualization of performance improvements following the hyperparameter tuning with SA and GA across various models and metrics. Notably, the ensemble model shows distinct enhancements when tuned with SA, achieving superior accuracy, precision, sensitivity, specificity, and F1-score compared to when tuned with GA. For instance, the SA-tuned ensemble achieved a sensitivity of 0.858491, noticeably higher than the 0.823944 seen with GA tuning. This trend is consistent with the improvements observed when comparing the untuned ensemble, which displayed uniform metrics across all categories at 0.816901, demonstrating substantial gains particularly in sensitivity and F1-score after tuning, highlighting the effectiveness of SA in balancing performance metrics.

In individual base classifiers, the ANN model's sensitivity improved significantly from 0.690141 in its untuned state to 0.774648 with SA, outperforming the improvement to 0.711268 with GA. This pattern indicates that SA is particularly effective for ANN in enhancing sensitivity. Conversely, XGBoost also benefitted from SA tuning, which elevated its sensitivity to 0.823944 compared to 0.802817 with GA, reinforcing SA's suitability for this model. Meanwhile, CatBoost's sensitivity enhanced from an initial 0.760563 to 0.795775 with SA and to 0.781690 with GA, with SA again proving more efficacious.

A closer examination of LDA reveals a slight decrease in sensitivity from 0.725352 to 0.711268 post-tuning, suggesting minimal impact from both SA and GA in this instance. However, across all models, SA generally yielded higher accuracy, precision, and notably, F1-scores, which represent the balance between precision and sensitivity. This is particularly crucial in clinical applications where accurately identifying patients needing ventilators is paramount to prevent adverse outcomes.

The overarching trends from this comprehensive evaluation clearly illustrate that hyperparameter tuning, especially with SA, significantly elevates the performance of ensemble and base classifiers in predicting ventilator requirements in the ICU, with SA typically outperforming GA in optimizing key metrics critical for clinical decision-making.

Table 7 compares the results of the untuned ensemble model, as shown in Table 4, with those of the ensemble models whose base classifiers were tuned using SA and GA, and also with an AutoML approach. This table illustrates that the SA-tuned ensemble achieved the highest scores in accuracy, precision, and F1-score, surpassing those of the GA-tuned ensemble and AutoML, indicating superior performance in balancing sensitivity and specificity.

The optimization plots for the ANN, LDA, CatBoost, and XGBoost models using both SA and GA, depicted in Figs 3 and 4, show that the optimization processes for both algorithms reach a steady state, ensuring that the accuracy indices remain stable and do not degrade over time. These figures provide a visual confirmation of the algorithms' efficacy in maintaining robust model performance throughout the tuning process.

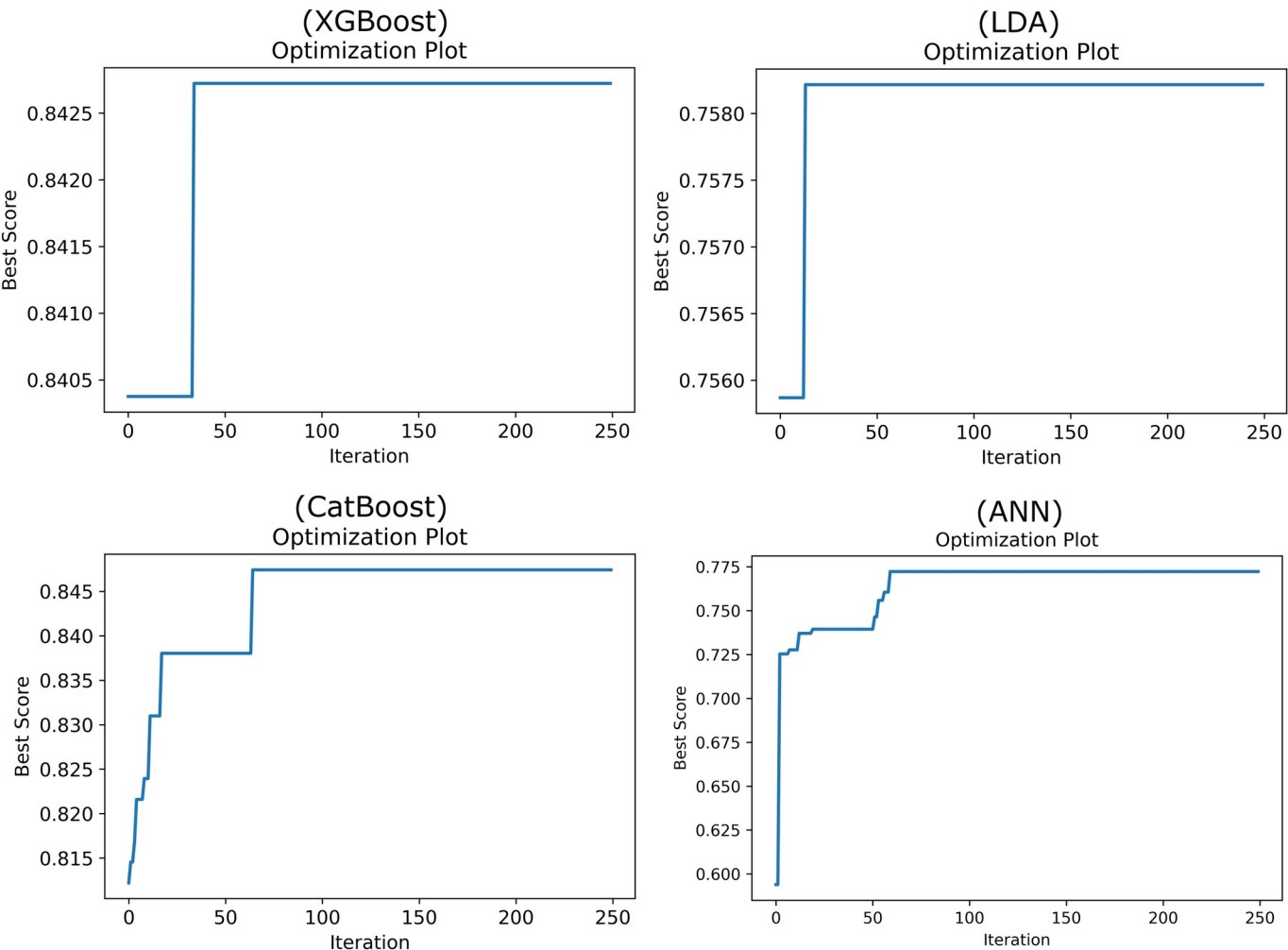

**Fig 3. Optimization plot of ML models using SA algorithm.**

Through a comprehensive evaluation, it was determined that the ensemble model that its base classifiers tuned with SA achieved the best overall performance in predicting ventilator requirements for patients in the cardiac surgery ICU, showcasing an exemplary use of hyper-parameter optimization to enhance model reliability and effectiveness.

As we conclude, we acknowledge that our research, which uses data from a single hospital due to data gathering limitations, might affect the generalizability of our findings. Despite employing data partitioning and 10-fold cross-validation to improve reliability, further external validation is necessary. Future studies will aim to incorporate data from multiple hospitals to bolster the robustness and applicability of our model in diverse clinical settings.

After handling missing values, we conduct correlation analysis on the features. We calculate Pearson's correlation coefficient for each feature against all other features [22]. The visualization of the correlation analysis is provided in Fig 5.

As shown in Fig 5, some features have a high correlation with the response variable (both positive and negative), such as the columns: "CABG_Valve", "Cross_clamp", "number_of_-grafts", "Age", "Weight", and etc. In related studies, the significance of selected features—such as "CABG_Valve" [64], "Cross_clamp" [65], "Age" [66] and "Weight" [67]—has been a topic

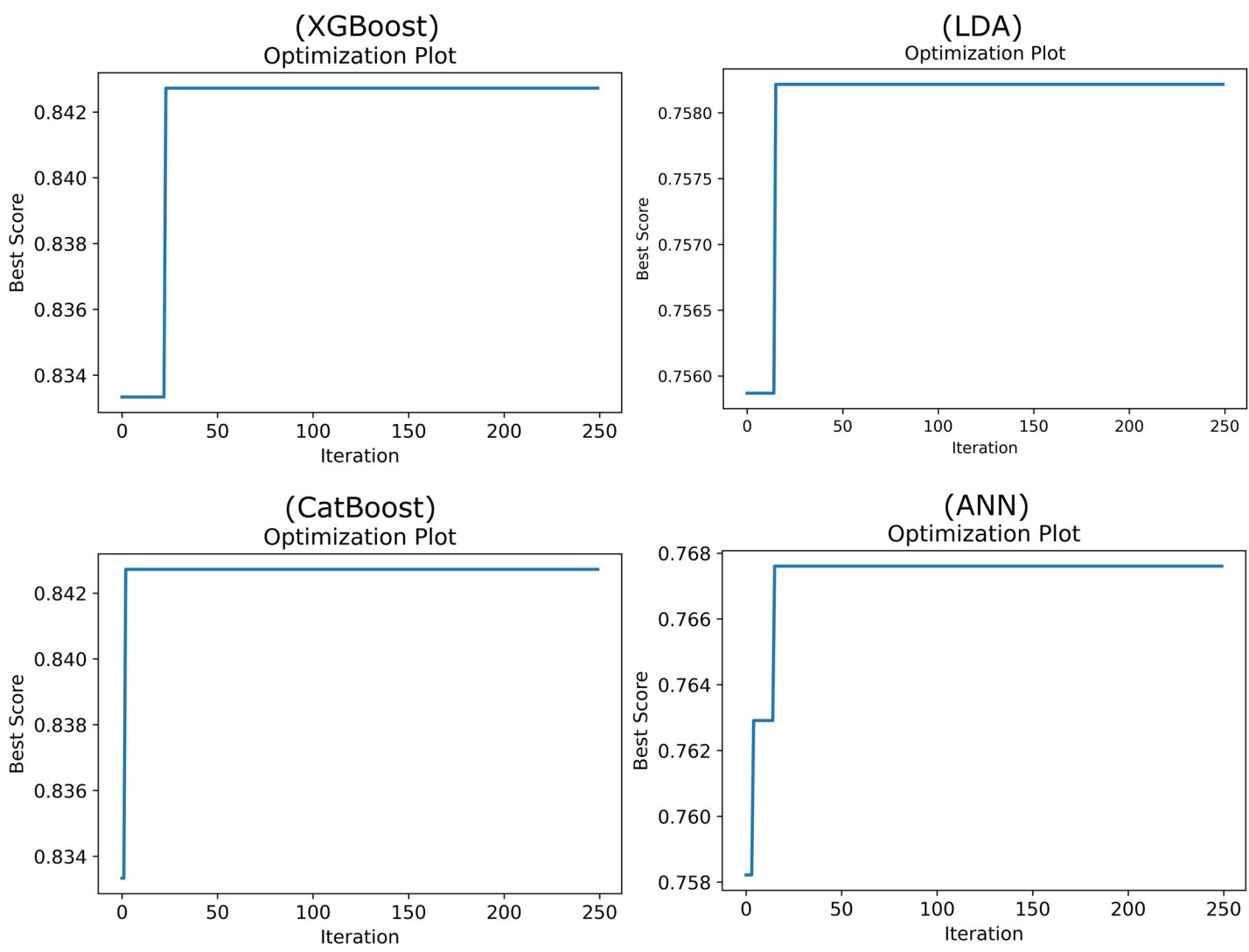

**Fig 4. Optimization plot of ML models using GA algorithm.**

of discussion. These features play a crucial role in various contexts, including cardiac surgery outcomes and patient care.

Also, when we compare these highly correlated variables with the results from the feature importance section (obtained by RF and GBM), it is clear that some features are also considered as highly important features in the feature importance results. This commonality between the results indicates that in addition to having a high correlation with the response variable, they have a significant effect in predicting the response variable. Therefore, it is important to know and pay attention to these important clinical features. For example, Visualizing data related to 'CABG_Valve' and 'Cross_clamp' (Fig 6) reveals a high proportion of ventilated patients in class 1 for these features. Also As observed in the Fig 7, with increasing age group, the EF trend decreases. In other words, based on this dataset and these graphs, it can be seen that the blood exchange between the atrium and ventricle of the heart decreases on average as the age group becomes higher.

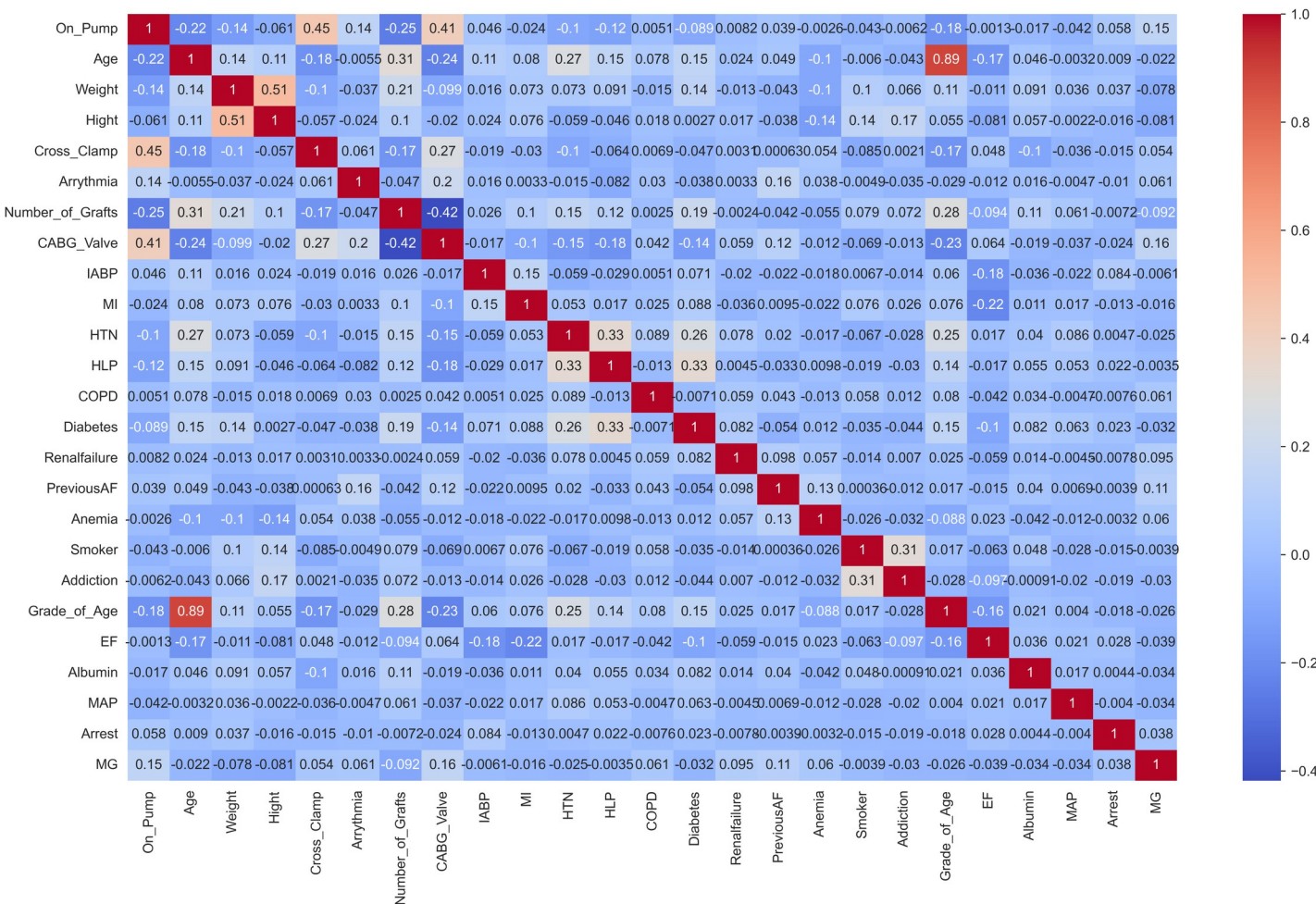

**Fig 5. Correlation analysis between features.**

This also shows the importance of these clinical features in the cardiac surgery ICU and can help clinical staff to pay attention more to this matter. Consequently, analyses like the findings obtained in this study can significantly impact patient care and the management of healthcare resources.

## 4. Conclusions

In the realm of healthcare, a substantial amount of data is being generated, offering a potential goldmine of insights that can be unearthed using ML and DL techniques. The cardiac surgery ICU holds a distinct position within hospitals, providing ample data for valuable analyses. This study introduces a hybrid model combining ML and DL with a metaheuristic algorithm for predicting ventilator needs in cardiac surgery ICUs, addressing the urgent need to manage limited ventilator resources and prevent the life-threatening and financial consequences of delayed ventilator allocation to critical patients. We propose this model as a timely solution to this critical challenge.

Our analysis revealed that the ensemble model, with its base classifiers tuned using the SA optimization algorithm, achieved superior performance compared to other models. Furthermore, the results indicated that the metaheuristic algorithm, such as SA and GA, effectively

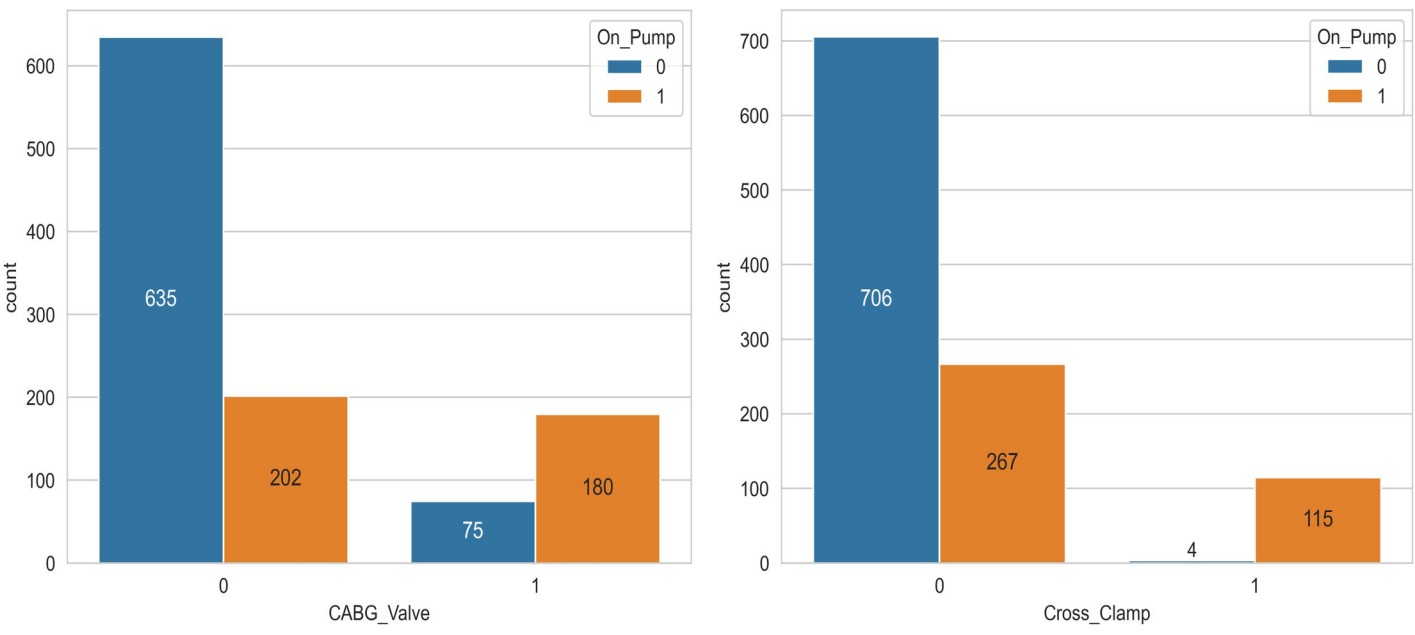

**Fig 6. Countplot of 'CABG_Valve' and 'Cross_Clamp' Based on 'On_Pump'.**

enhanced the model's predictive performance. By leveraging these techniques, we can effectively distinguish between patients admitted to the cardiac surgery ICU who require ventilator assistance and those who do not, enabling medical staff to make informed decisions in critical situations, balancing resource constraints and patient condition.

However, our study does have limitations. The dataset used is from a single center, which may limit the generalizability of our findings. Future work should include validating our

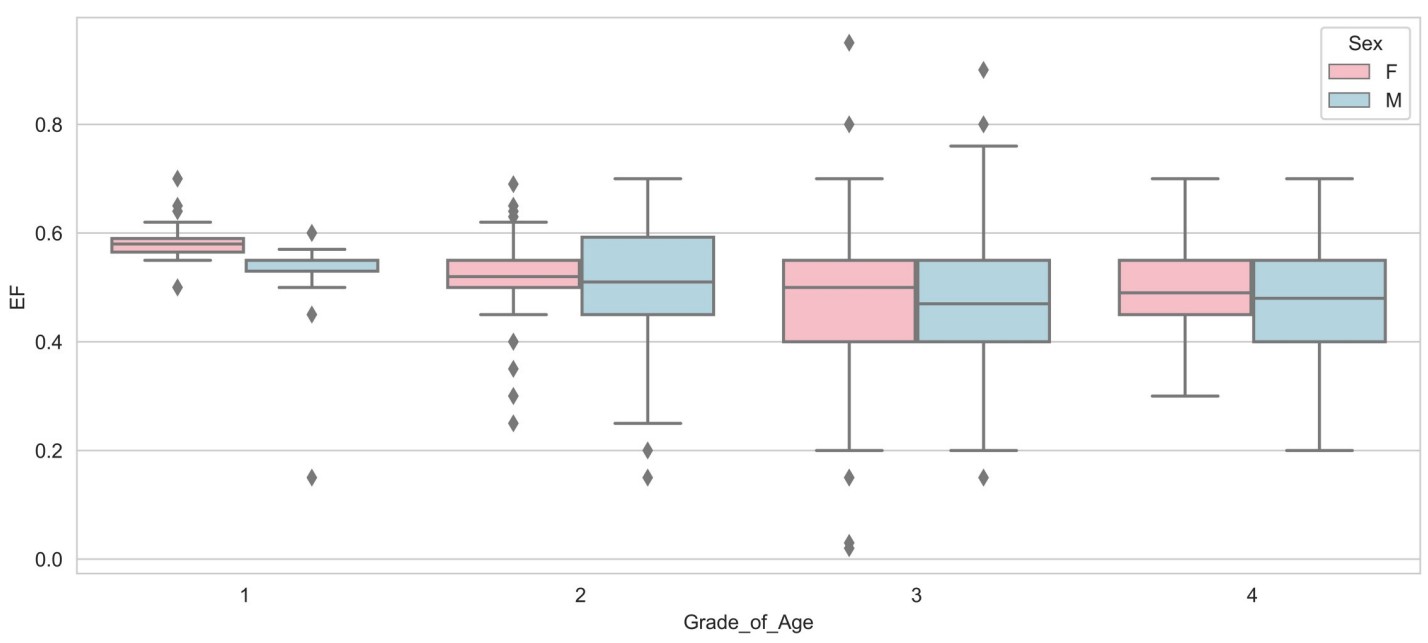

**Fig 7. Boxplot of 'EF' versus 'Grade of Age' Stratified by 'Sex'.**

model on multi-center datasets to ensure broader applicability. Additionally, further research could explore the use of alternative metaheuristic algorithms or innovative approaches to optimize the hyperparameters of ML and DL models for improved evaluation metrics. Exploring and incorporating alternative feature selection methods could also further optimize the model's performance.

## Author Contributions

**Conceptualization:** Ali Bahrami, Morteza Rakhshaninejad, Rouzbeh Ghousi, Alireza Atashi.

**Data curation:** Ali Bahrami, Alireza Atashi.

**Methodology:** Ali Bahrami, Morteza Rakhshaninejad, Rouzbeh Ghousi, Alireza Atashi.

**Resources:** Alireza Atashi.

**Software:** Ali Bahrami, Morteza Rakhshaninejad.

**Supervision:** Rouzbeh Ghousi.

**Validation:** Ali Bahrami, Alireza Atashi.

**Visualization:** Ali Bahrami, Morteza Rakhshaninejad.

**Writing – original draft:** Ali Bahrami, Morteza Rakhshaninejad.

**Writing – review & editing:** Rouzbeh Ghousi, Alireza Atashi.

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
