## [Decision Letter · Decision Letter 0]

16 May 2024

PONE-D-24-07915Enhancing Machine Learning Performance in Cardiac Surgery ICU: Hyperparameter Optimization with Metaheuristic AlgorithmPLOS ONE

Dear Dr. Ghousi,

Thank you for submitting your manuscript to PLOS ONE. After careful consideration, we feel that it has merit but does not fully meet PLOS ONE’s publication criteria as it currently stands. Therefore, we invite you to submit a revised version of the manuscript that addresses the points raised during the review process.

We look forward to receiving your revised manuscript.

Kind regards,

Xin Gao

Academic Editor

PLOS ONE

Additional Editor Comments:

Dear Author(s),

Thank you for submitting your manuscript to PLOS ONE. We appreciate the effort you have put into this research and the potential impact it can have on ICU ventilator management. After carefully reviewing the feedback from the reviewers, I have decided that a major revision is necessary before we can consider your manuscript for publication.

Given the mixed reviews, I have decided that a major revision is necessary. Please address the following key points to improve your manuscript:

Provide clear explanations for your choice of models (CatBoost, LDA, ANN) and the use of simulated annealing over other methods such as cross-validation or grid search. Additionally, ensure that the rationale behind your hyperparameter tuning ranges is well-articulated.

Reconsider the approach of removing features with more than 15% missing data. Explore and discuss imputation methods and assess the importance of these features to ensure they are not prematurely discarded.

Validate your model across multiple independent datasets to enhance its robustness and generalizability. Address potential overfitting issues, particularly with ANN, and consider the use of ensemble models for improved predictive performance.

(Optional) Include comparisons of simulated annealing with other optimization algorithms and techniques such as AutoML. This will help demonstrate the advantages and limitations of your approach.

Discuss the clinical significance of your findings in more detail and incorporate aspects of model explainability to improve trust and adoption in clinical settings. Highlight how your model integrates with existing scoring systems and its potential impact on patient care.

Ensure that all figures and plots in the manuscript are of high quality and clearly legible.

Please provide a detailed response to the reviewers' and editor's comments and submit a revised version of your manuscript along with a point-by-point response letter. The revised manuscript will undergo another round of review to ensure the concerns have been adequately addressed.

We look forward to receiving your revised manuscript and hope that you can address the issues raised to improve the quality and impact of your study.

Best regards,

Reviewers' comments:

Reviewer's Responses to Questions

**Comments to the Author**

1. Is the manuscript technically sound, and do the data support the conclusions?

Reviewer #1: Yes

Reviewer #2: Yes

Reviewer #3: Partly

Reviewer #4: Yes

Reviewer #5: No

Reviewer #6: Yes

2. Has the statistical analysis been performed appropriately and rigorously? 

Reviewer #1: Yes

Reviewer #2: Yes

Reviewer #3: N/A

Reviewer #4: Yes

Reviewer #5: Yes

Reviewer #6: Yes

3. Have the authors made all data underlying the findings in their manuscript fully available?

Reviewer #1: Yes

Reviewer #2: Yes

Reviewer #3: Yes

Reviewer #4: No

Reviewer #5: No

Reviewer #6: Yes

4. Is the manuscript presented in an intelligible fashion and written in standard English?

Reviewer #1: Yes

Reviewer #2: Yes

Reviewer #3: Yes

Reviewer #4: Yes

Reviewer #5: Yes

Reviewer #6: Yes

5. Review Comments to the Author

Reviewer #1: it is a well-designed and executed study that addresses an important challenge in the healthcare domain. The combination of ML models with SA for hyperparameter optimization is an effective approach that has not been widely explored in the context of cardiac surgery ICU ventilator prediction. The enhancement in the predictive performance, particularly the increase in sensitivity (recall), after applying the Simulated Annealing algorithm for hyperparameter tuning is a noteworthy finding that can have direct implications for patient care and resource allocation in the ICU setting.

To further strengthen your manuscript, I would suggest considering discussing the clinical significance of your findings in more detail, highlighting the potential impact on patient care and healthcare resource management. Also acknowledge the limitations of your study, such as the single-center nature of the dataset, and discuss the need for external validation of the proposed models. Lastly, explore the possibility of integrating your hybrid model with existing scoring systems (e.g., APACHE, SAPS) to further enhance the predictive capabilities and provide a more comprehensive assessment of patient risk.

Reviewer #2: In the paper titled "Enhancing Machine Learning Performance in Cardiac Surgery ICU: Hyperparameter Optimization with Metaheuristic Algorithm," the authors present a significant advancement in predictive healthcare analytics. The study addresses the critical need for ventilator allocation in cardiac surgery ICUs, leveraging a hybrid approach that combines machine learning (ML) and deep learning (DL) models with a metaheuristic algorithm.

One of the key strengths of this study lies in its innovative methodology, which employs Simulated Annealing (SA) for hyperparameter tuning, significantly improving the prediction performance of ML and DL models. Notably, the study demonstrates a remarkable 14% increase in sensitivity for Artificial Neural Networks (ANN) after applying SA.

Furthermore, the authors provide comprehensive insights into the challenges of ICU care, particularly in ventilator management, and highlight the potential of ML techniques to address these challenges effectively. By leveraging a rich dataset obtained from a hospital, the study offers practical implications for optimizing ventilator allocation in critical healthcare settings.

The paper's contribution extends beyond methodology, as it systematically addresses the gap in predicting ventilator needs specifically for cardiac surgery ICU patients. This focus is particularly relevant during pandemics or periods of increased ICU admissions, where efficient resource allocation can be lifesaving.

Overall, the study provides a well-researched and meticulously executed analysis that not only enhances our understanding of ventilator management in cardiac surgery ICUs but also underscores the potential of ML techniques, especially when integrated with metaheuristic algorithms, to improve patient care outcomes in critical healthcare scenarios.

Reviewer #3: Thanks for your contribution. I have couple questions below:

Technical:

1. The paper conducted PCA but the feature space is not large - 32 features and PCA can throw unsatisfying results. Normally L1, L2 regularization can alleviate overfitting issue, all of the LDA/Catboost/ANN have ways to do that.

2. Do have concern over removing missing feature > 15%. Some may be important and useful in prediction. Have we tried any imputation method for missing values. Probably looking at feature importance to see if some missing features are badly removed.

3. The paper experimented with CatBoost/LDA/ANN but didn't give articulate the reasons behind it.

4. The paper didn't articulate the reasons using SA to conduct hyper-parameter tunings although its valid theoretically. Other methods, e.g. Cross-validation can do the tuning easily.

5. Have we checked overfitting issues when evaluation the models, ANN can be easily overfitted.

6. For predictive purpose, it may be benefitted from an ensembled model.

7. The paper lacks comparison of SA to other optimization algorithm.

Publication:

The plots in the end seem blurred.

Reviewer #4: This study introduces a novel approach to predict the need for ventilators in cardiac surgery ICU patients by integrating machine learning (ML) and deep learning (DL) models with a metaheuristic algorithm, specifically Simulated Annealing (SA), for hyperparameter optimization. The research is highly relevant and valuable in reality, given the critical importance of ventilator allocation in healthcare settings, especially during the pandemic.

Strengths:

1. The paper effectively combines various ML and DL techniques with a metaheuristic optimization algorithm, which is innovative in the field of medical data.

2. The topic is of high relevance, potentially offering significant improvements in ICU resource management and patient care.

Weakness:

1. The study is limited to data from a single hospital source and the data access is restricted. Validation of the model across multiple independent datasets would enhance the robustness and general applicability of the findings.

2. There is insufficient discussion regarding the limitations of the applied models and the potential biases within the study. A thorough exploration of these aspects would provide a more comprehensive understanding of the model's utility and constraints.

3. In clinical settings, the explainability of predictive models is crucial for user trust and adoption. The study could be improved by incorporating aspects of model explainability and interpretability, particularly in explaining decisions in critical care scenarios.

Reviewer #5: The problem stated in this paper is relatively simple in domain of machine learning. I did not deny its importance in the field of medical domain, but such a classification/regression problem (given data of a patient, determine if he/she needs ventilator) has been well studied, and many advanced techniques can be used to address this problem. Applying traditional machine learning algorithm such LDA, Catboost and ANN is nothing new. More advanced research has been using tabular transformers to address medical data like this.

Regarding the Simulated Annealing (SA) to improve the ML and DL models' performance. I think a more realistic way of doing hyper parameter is doing grid search. The claimed performance improvement is questionable given the size of the dataset used in this paper (1092 records in total). The difference between each method, and before and after using SA is minor. Plus, i think in medical domain, sensitivity is more important than other metrics. assigning ventilator to a patient who does not need it might be a waster of resource, but not assigning a ventilator to a patient who actually need it will be life threatening consequence. If we take a look at the sensitivity of all model results provided in this paper, it's around 50%, which is more like a random guessing given this is a binary classification problem (need or not need a ventilator).

Reviewer #6: Technically, I think this paper is well written. The algorithms and methodology are introduced clearly. I do have several problems from my perspective:

1. I think we do need some intuition of using certain ML mythology. Table1 introduces some of the ML method that other paper were using. It seems you are using some method that others not used before. But that is not sufficient to demonstrate the reason us using CatBoost, LDA or ANN.

2. If the focus for this paper is hyperparameter tuning, I think you need to explain why you choose such ranges for parameter tuning.

3. If we also want to compare between different models, we might want to make sure their search space are similar, or at least within the similar number of parameter combinations. It seems ANN's search space is bigger than other two. To this end, I think the comparison might not be fair.

4. There are many hyperparameter tuning methods. For example, AutoML works well if you want to compare between models. Can you compare the pros and cons of your method versus AutoML frameworks? In other words, why not using AutoML.

6. PLOS authors have the option to publish the peer review history of their article (what does this mean?). If published, this will include your full peer review and any attached files.

Reviewer #1: No

Reviewer #2: **Yes: **Yuhe Zhang

Reviewer #3: No

Reviewer #4: No

Reviewer #5: No

Reviewer #6: No

---

## [Author Response · Author response to Decision Letter 0]

4 Jul 2024

Dear Editor and Reviewers,

Thank you for your insightful feedback on our research. We have diligently addressed each of your comments and made the necessary revisions accordingly.

Please find attached files in the editorial manager system the documents including our Response to Reviewers, the Revised Manuscript with Track Changes, and the Clean Version of the Revised Manuscript for your convenience.

We appreciate your time and look forward to your continued guidance.

Best regards, 

Rouzbeh Ghousi

---

## [Decision Letter · Decision Letter 1]

17 Sep 2024

Enhancing Machine Learning Performance in Cardiac Surgery ICU: Hyperparameter Optimization with Metaheuristic Algorithm

PONE-D-24-07915R1

Dear Dr. Ghousi,

We’re pleased to inform you that your manuscript has been judged scientifically suitable for publication and will be formally accepted for publication once it meets all outstanding technical requirements.

Kind regards,

Xin Gao

Academic Editor

PLOS ONE

Additional Editor Comments (optional):

Dear Authors,

I am pleased to inform you that your manuscript, "Enhancing Machine Learning Performance in Cardiac Surgery ICU: Hyperparameter Optimization with Metaheuristic Algorithm," has been accepted for publication in PLOS ONE.

The reviewers and I appreciate the efforts you have made in addressing the concerns raised during the review process. Your revisions have significantly improved the quality and rigor of your work, and we believe your research makes a valuable contribution to the field of machine learning applications in cardiac surgery ICU ventilator management.

Specifically, you have:

1. Provided a more comprehensive discussion on the limitations of using a single-center dataset and the need for external validation using multi-center datasets to enhance generalizability.

2. Clarified your feature selection process, including justifications for the chosen scaling and upsampling methods.

3. Explained your cross-validation strategy and its importance given the sample size.

4. Corrected the reported metrics for the ensemble model, ensuring accuracy.

5. Strengthened the rationale for selecting specific machine learning models and their suitability for the classification problem at hand.

6. Addressed other specific concerns raised by the reviewers, improving the overall clarity and quality of the manuscript.

However, there are some concern that should be addressed: please double-check the reported metrics for the ensemble model, as it is unlikely for all metrics (accuracy, precision, sensitivity, specificity, and F1-score) to have the same value. Ensure the accuracy of the reported results.

While you have discussed the need for external validation, your study is still limited to a single-center dataset. In your future research, we strongly encourage you to prioritize multi-center validation to confirm the generalizability of your findings.

We believe your work represents a significant contribution to the field and has the potential to inform clinical practice and inspire further exploration in this area.

On behalf of the editorial team, I would like to congratulate you on your achievement and thank you for your dedication to this research. We look forward to the publication of your manuscript in PLOS ONE.

Best regards,

Reviewers' comments:

Reviewer's Responses to Questions

**Comments to the Author**

1. If the authors have adequately addressed your comments raised in a previous round of review and you feel that this manuscript is now acceptable for publication, you may indicate that here to bypass the “Comments to the Author” section, enter your conflict of interest statement in the “Confidential to Editor” section, and submit your "Accept" recommendation.

Reviewer #4: All comments have been addressed

Reviewer #7: (No Response)

2. Is the manuscript technically sound, and do the data support the conclusions?

Reviewer #4: Yes

Reviewer #7: Partly

3. Has the statistical analysis been performed appropriately and rigorously? 

Reviewer #4: Yes

Reviewer #7: Yes

4. Have the authors made all data underlying the findings in their manuscript fully available?

Reviewer #4: No

Reviewer #7: Yes

5. Is the manuscript presented in an intelligible fashion and written in standard English?

Reviewer #4: Yes

Reviewer #7: Yes

6. Review Comments to the Author

Reviewer #4: Thank you for addressing my comments with detailed responses and revisions.

Dataset Limitation: Your internal validation methods, including the use of ensemble learning and 10-fold cross-validation, are commendable and should strengthen the reliability of your findings. However, I still encourage future research to explore multi-center datasets when feasible to enhance generalizability.

Model Limitations: Your explanation of the ensemble method and its impact on reducing overfitting and bias is well-articulated.

Explainability: The integration of model explainability techniques, such as feature importance analysis, is a positive development. This addition should indeed help in bridging the gap between complex analytics and practical clinical application.

Reviewer #7: The paper has proposed a new approach, which is ML-based to optimize ventilator allocation in cardiac surgery ICUs. The use of metaheuristic algorithms for hyperparameter tuning adds value to the existing body of research. Regarding the ML part of the paper, it explored multiple ML and deep learning models, including LDA, CatBoost, ANN, and XGboost, and used the weight-voting mechanism for the models. The paper is well-written

Recognizing the paper's contribution to the prediction for surgery ICUs I do have some questions to this paper.

- On page 7 regarding the feature selection, first, regarding the ID for each patient, if there’s no specific meaning of the ID, instead just like a random number, it might not be necessary to use that as one feature. Meanwhile, regarding the standardization of the numeric data, before doing the z-score scaling, it would be better to explore the data distribution first to show that z-score scaling is the most appropriate scaling method for the chosen datasets. This also applies to why use SMOTE to do the upsampling instead of using other up-sampling methods.

- Some parts of the paper are lengthy, for example like the parts that explains the reason to split the dataset into train, validation, and test and the importance of validation datasets. Meanwhile, I wonder is there any cross-validation strategy used in splitting the datasets. Because the data sample size is kind of small, it might be better to apply some similar strategy to do data augmentation.

- For table 4 ensemble model results, all metrics (Accuracy, precision, sensitivity, specificity and F1-Score) have given the same results 0.8169, wonder if there’s any error in this result because it is not likely that the model will have the same number across different metrics.

- Regarding the ML technology is using (CatBoost, LDA, ANN and XGBoost), as the ML problem basically falls into classification, I wonder if there’s a particular reason for picking ANN and these boosting models. There could be choices of other models, and I feel the paper didn’t give strong reason or motivation for picking up these method.

7. PLOS authors have the option to publish the peer review history of their article (what does this mean?). If published, this will include your full peer review and any attached files.

Reviewer #4: No

Reviewer #7: No

---

## [Editor Report · Acceptance letter]

22 Oct 2024

PONE-D-24-07915R1 

PLOS ONE

Dear Dr. Ghousi, 

I'm pleased to inform you that your manuscript has been deemed suitable for publication in PLOS ONE. Congratulations! Your manuscript is now being handed over to our production team.

Kind regards, 

on behalf of

Dr. Xin Gao 

Academic Editor

PLOS ONE